# Long-term patterns of gender imbalance in an industry without ability or level of interest differences

**Luís A. Nunes Amaral**[1,2,3]*, **João A. G. Moreira**[2], **Murielle L. Dunand**[1], **Heliodoro Tejedor Navarro**[1], **Hyojun Ada Lee**[2]

**1** Northwestern Institute on Complex Systems, Northwestern University, Evanston, IL, United States of America, **2** Dept. Chemical and Biological Engineering, Northwestern University, Evanston, IL, United States of America, **3** Dept. Physics and Astronomy, Northwestern University, Evanston, IL, United States of America

\* amaral@northwestern.edu

**Data Availability Statement:** The datasets analyzed in this study are available as Supporting Information files and at https://arch.library.northwestern.edu.

## Abstract

Female representation has been slowly but steadily increasing in many sectors of society. One sector where one would expect to see gender parity is the movie industry, yet the representation of females in most functions within the U.S. movie industry remain surprisingly low. Here, we study the historical patterns of female representation among actors, directors, and producers in an attempt to gain insights into the possible causes of the lack of gender parity in the industry. Our analyses reveals a remarkable temporal coincidence between the collapse in female representation across all functions and the advent of the Studio System, a period when the major Hollywood studios controlled all aspects of the industry. Female representation among actors, directors, producers and writers dropped to extraordinarily low values during the emergence and consolidation of the Studio System that in some cases have not yet recovered to pre-Studio System levels. In order to explore some possible mechanisms behind these patterns, we investigate the association between the gender balance of actors, writers, directors, and producers and a number of economic indicators, movie industry indicators, and movie characteristics. We find robust, strong, and significant associations which are consistent with an important role for the gender of decision makers on the gender balance of other industry functions. While in no way demonstrating causality, our findings add new perspectives to the discussions of the reasons for female under-representation in fields such as computer science and medicine, that have also experienced dramatic changes in female representation.

## Introduction

Gender diversity is increasingly regarded as a desirable condition by educational, business, and governmental organizations. Recent research shows that more gender-balanced groups are better at complex decision-making [1] and that females show less self-interest and are better at complex moral reasoning than males. [2] Indeed, the proportion of women faculty

**Funding:** JAGM thanks Fundação para a Ciência e Tecnologia for grant SFRH-BD-76115-2011, LANA thanks Department of Defense's Army Research Office for grant 281 W911NF-14-1-0259, the John Templeton Foundation for Award FP053369-A// 39147 and the John and Leslie McQuown Gift.

**Competing interests:** The authors have declared that no competing interests exist.

members in many STEM fields has been steadily increasing, [3] as has the number of females in corporate suites and in political office. [4, 5] These trends are significant because the absence of women in leadership positions was reported to have a negative impact on women's aspirations and advancement and may perpetuate gender biases. [6, 7]

Still, most fields struggle to reach gender parity. A particularly striking example is the computer software industry. [8–11] While nowadays female representation is remarkably low, there is clear evidence of early contributions by women to computer science. The first recorded computer algorithm was written by Ada Lovelace. During World War II, female mathematicians programmed the computers used for code-breaking at Bletchley Park in Britain, and U.S. female mathematicians programmed the ENIAC and the UNIVAC. The first modern computer language and compiler were developed by Grace Hopper, while another female programmer, Evelyn Berezin, developed the first mainstream word processor and the world's first computerized airline reservation system. In fact, until the 1970's coding was seen as a women's job, and as recently as the mid 1980's, nearly 40% of computer science graduates were female. [8, 10] Nowadays, that number is only about 17% [9] and there is an ongoing controversy within the industry on whether females have an interest and/or innate ability for coding.

The movie industry, in contrast to the software industry, is a case where some of the mechanisms that are sometimes hypothesized to drive gender differences are unlikely to operate. For example, in spite of the contributions of female coders, most people are better able to recognize the prominent men in the software industry than the prominent women—consider the name recognition of Lovelace, Hopper, and Berezin, versus that of Turing, Gates and Zuckerberg. The situation is quite different for acting. While it took until the 1600's for women to be allowed on the stage, they quickly proved their singing and acting talent in operas and plays. Already by 1869, only two centuries after they were first allowed on stage in England, John Stuart Mill and Harriet Taylor, would point out that "The only one of the fine arts which women do follow, to any extent, as a profession, and an occupation for life, is the histrionic; and in that they are confessedly equal, if not superior, to men." [12] Marlene Dietrich, Katharine Hepburn, and Meryl Streep are just as recognizable and easy to name as Douglas Fairbanks, Humphrey Bogart, and Tom Hanks. Moreover, because the time required of most actors to be on set during the filming of a movie is quite brief compared with most other jobs, film acting as a career significantly reduces the possible career impact of maternity leave and childrearing. [13] Yet, there is evidence of significant gender imbalances in U.S.-produced movie casts. [6, 14–19]

We study the temporal evolution of female representation in the movie-making teams of over 26 thousand United States (US) produced movies released between 1911 and 2010 (Fig 1; see Methods for details on criteria for inclusion). We find a striking temporal coincidence between the creation and consolidation of the Hollywood Studio System [20, 21] and a drastic reduction in gender balance across several functions in the US movie industry. Additionally, we find that the dearth of female representation among movie producers during the heydays of the Studio System is strongly associated with lower lower levels of female representation among directors, screenwriters, and actors. We also find an intriguing temporal coincidence between the breakup of the Studio System—and the consequent increase in the bargaining power of actors, both male and female—and the increased inclusion of former actors among the producer ranks.

The implications of our findings extend far beyond the movie industry. Indeed, while the study of the historical patterns of female representation among actors and other movie-making functions is a very important field of study on its own, our perspective is that our study is likely to also contribute significant new insights into the reasons why females are under-represented

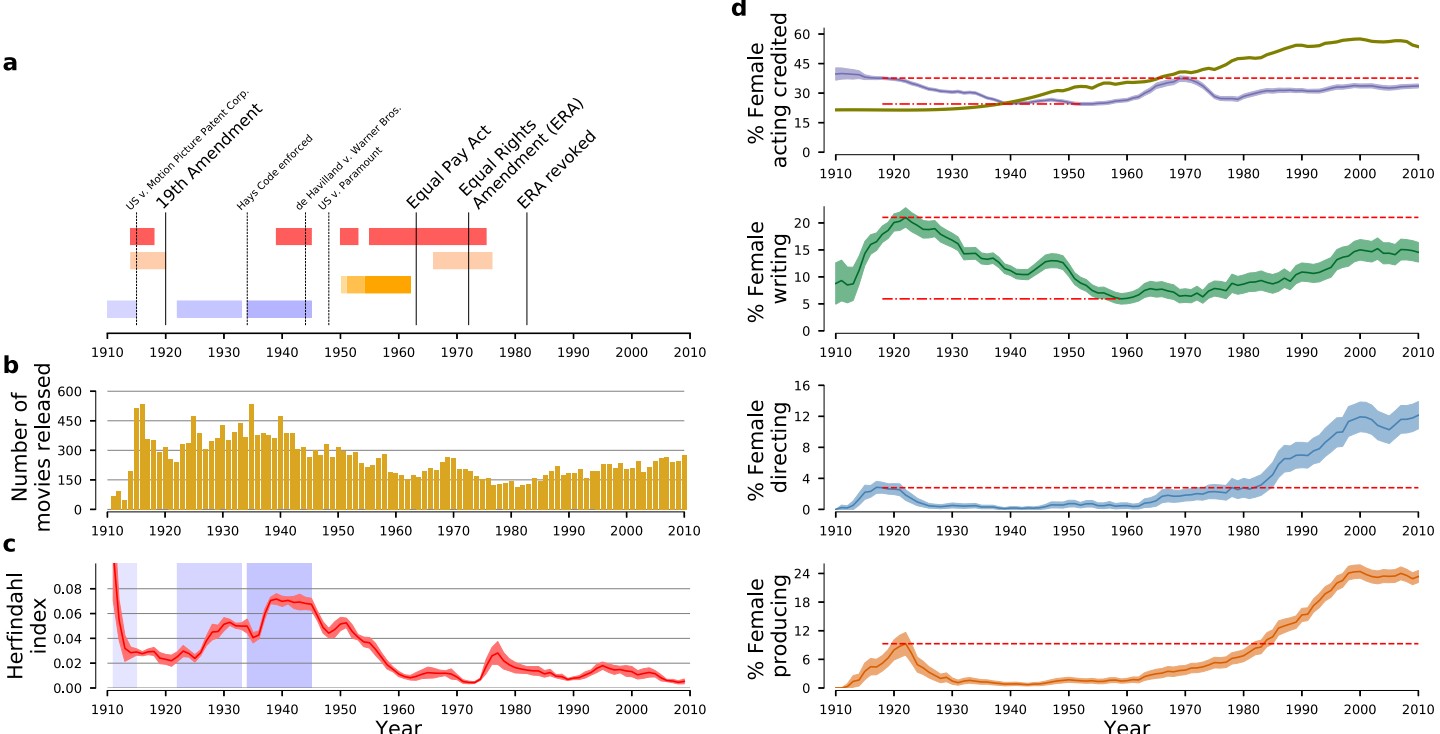

**Fig 1. Historical trends of gender imbalance in the U.S. movie industry.** (**a**) Timeline of 20th century events relevant to the evolution of the U.S. movie industry. Red bars identify major wars with U.S. involvement (chronologically, World War I, World War II, Korean war, and Vietnam war). Light peach bars show the peak of the feminism waves in the U.S. Orange shadings indicate the rates of TV adoption in U.S. households (from unsaturated to saturated: <30%, <60%, <90%). [47] Light blue shadings identify, from less to more saturated, MPPC control, [48] consolidation of the Hollywood Studio System, and Studio System's heyday. [20] The de Havilland vs. Warner Bros' 1944 decision freed major stars from life long contracts with as single studio. The U.S. vs. Paramount's 1948 decision unravelled the vertical integration of the U.S. movie industry. These legal defeats enabled greater negotiating power for movie stars and the ability of independent producing companies to have their movies distributed in a more equitable manner. (**b**) Number of U.S.-produced movies considered in our study released annually. See S1 Fig for breakdown by movie genre. (**c**) Concentration of the industry's output by producing company/studio as measured by the Herfindahl-Hirschman index. Larger values indicate that a large fraction of movies are produced by a small number of studios. We observe an increase of the degree of concentration during the period of control of the industry by MPPC. The degree of concentration reaches a minimum in 1922 but then it grows until 1944. These trends are consistent with current understanding of the consolidation and golden age of the Hollywood Studio System. [20, 48] (**d**) Temporal dependence of the percentage of females according to movie-making function. The olive solid line in the top panel shows female participation in the U.S workforce in order to provide a comparison to the overall economy. Solid lines indicate average values and the color bands show 95% confidence intervals. The red dotted lines show the highest value of female representation attained prior to 1940, and the red dashed lines show the lowest value attained after 1930, if different from zero. For all movie-making functions there is a similar "U-shape" pattern with an early maximum achieved prior to 1922, a minimum reached by the mid-1940s, and an increase after.

in, for example, STEM fields, without the confounding effect of *hypothetic* lower female innate abilities or levels of interest for the profession.

Indeed, there is currently a great deal of controversy [22] surrounding the primacy of evolutionary/biological [23, 24] mechanisms versus environmental/cultural mechanisms in explaining observed gender representation imbalances. We pursue here an approach informed by cultural primacy approaches. We believe this to be warranted since studies informed by evolutionary psychology suggest that women have a preference for activities where people and inter-personal interactions are prominent. [25] Such preferences would predict greater female interest for activities such as acting and producing. This hypothesis is supported by a recent survey reporting that high-school aged girls are more interested in careers in the arts, including acting, than males. [26]

If the gender imbalance observed in the movie industry would be due to cultural factors, then one needs to understand the reasons for discrimination. [27–32] Note that, in contrast to

Fleck and Hanssen, [33] we do no address here age discrimination, the interplay between age and gender, or the reasons why male stars are often cast to play roles younger than their chronological age. Rudman *et al.* [32] hypothesize that females who achieve prominence may ultimately experience a backlash because of status incongruity. This hypothesis is remarkably consistent with the historical trends we uncover. Moreover, backlash mechanisms for enforcing discrimination—such as the "boys club," and "locker room" culture [34–36]—are quite prominent in the context of the movie industry. The latter, expressed through references to the "casting coach", has become a trope of the movie industry and is described in many an actress memoir.

Following Becker [37], the question, nonetheless, remains of why would Hollywood's powers that be, accept the loss of profit arising from discriminating against talent? Even not accounting for those like Hitchcock who remarked that "actors are cattle", [36] a plausible answer is that it is hard to *a priori* predict the success of a movie project. [38, 39] Thus, it would have been quite easy to rationalize any losses arising from discriminating against talented females. In fact, even in fields where recognizing excellence earlier is easier, such as sports, racial discrimination was maintained for many decades. [40]

## Data

We retrieved release year, title, director, producers, writers, cinematographers, and production companies for 35,281 U.S.-produced movies compiled by the American Film Institute (AFI, https://search.proquest.com/afi/). Because AFI does not keep disambiguated information about industry participants, we also retrieved movie data from the Internet Movie Database (IMDb, https://imdb.com). We obtained release year, title, director, producers, writers, cinematographers, casting directors, and budget for 393,575 movies, and name, gender, birth year, and death year for 1,856,569 industry participants. Using titles, years of release and director names, we matched 25,645 records in the AFI database to movies in IMDb. These are the movies included in our study (see Methods for details).

## Results

### Century long trends in female representation

The emergence and consolidation of the Hollywood Studio System coincided with a dramatic change in the gender balance of a movie's cast (Fig 1 and Methods). Between 1922 and 1950, we observe a reduction of nearly 25% in the fraction of females in the cast of the typical movie. Surprisingly, not even the U.S.'s participation in World War II reversed this trend. In order to determine whether this decrease was due to changes in popularity of different movie genres (S1 Fig), we determine female representation in the cast of movies according to genre (Fig 2). While one does find that genres popular during this time period, such as Action, Adventure, and War, have smaller fractions of female cast members [17, 41], *the data leaves no doubt that the overall pattern of gender representation over time is replicated across all genres* (Fig 2).

The trends we report here for movie casts mirror trends reported earlier for screenwriters, [42] which we confirm (Fig 1d). The fraction of female writers steadily increased up to 1922, the year when industry concentration was lowest. Between 1922 and 1940, the fraction of female writers fell to half of its peak. The falling female representation continued until 1960, and has barely recovered above the 1940s level. [43, 44]

The temporal coincidence between the changes in female representation and the emergence and consolidation of the Studio System is not restricted to actors and writers. For both directors and producers we find similar—but even more extreme—patterns (Fig 1d). By the 1930s,

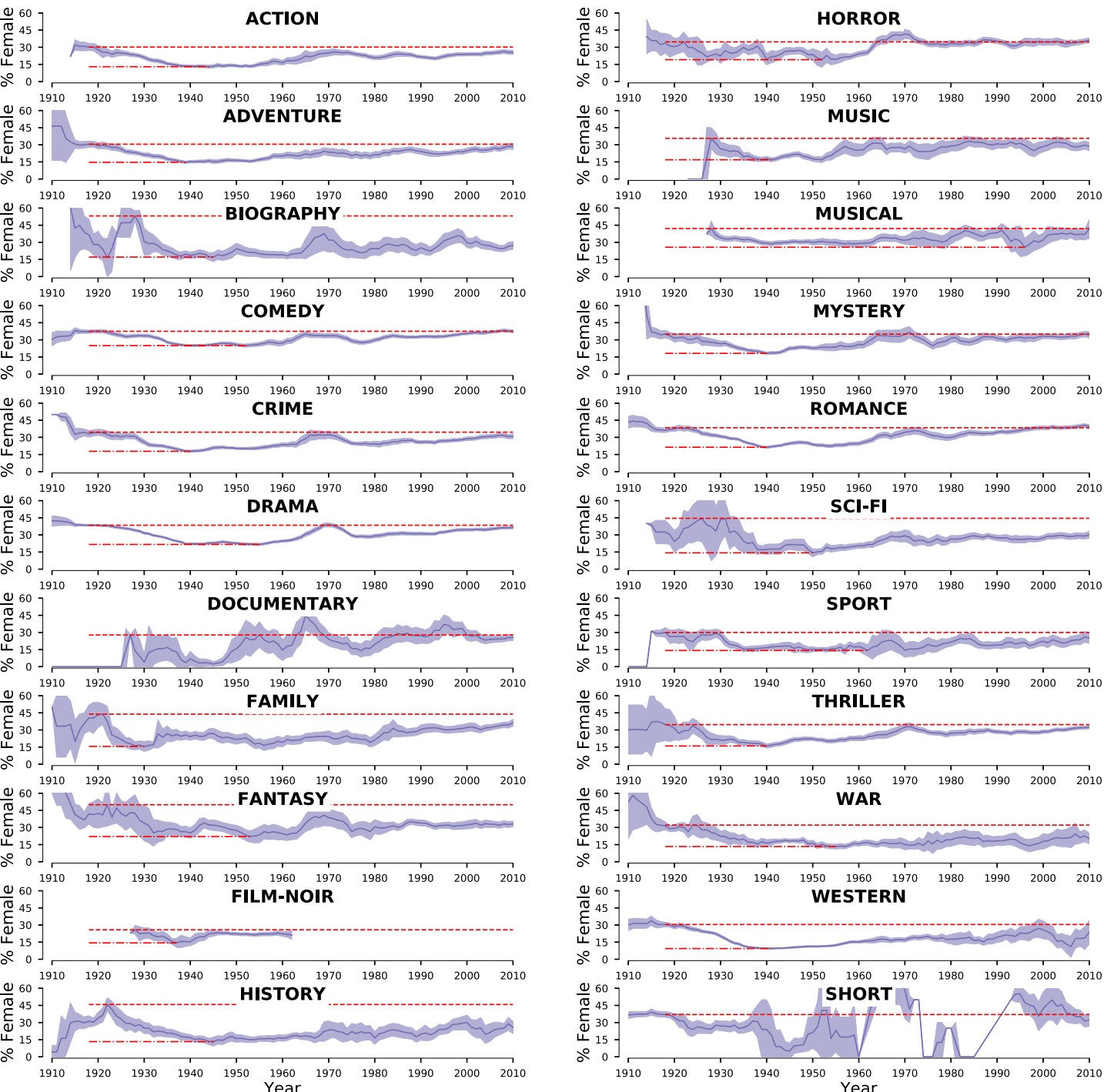

**Fig 2. Historical trends of gender imbalance in the casts of U.S.-produced movies.** Temporal dependence of the percentage of females actors according to movie genre. Solid lines indicate average values and the color bands show 95% confidence intervals. The red dotted lines highlight the highest value of female representation attained prior to 1940, whereas the red dashed lines highlight the lowest value attained after 1930. For all genres with sufficient number of movies, we find a clear "U-shape" pattern.

female representation among producers and directors had dropped to essentially zero. This collapse is particularly significant because producers' decisions are so important. [45, 46]

## Predictors of the level of female representation

In order to test for the presence of an association between industry consolidation and female representation, we conduct a multivariate regression of female representation in terms of factors capturing national economic indicators, movie industry indicators, and movie characteristics. Specifically, we consider the model:

$$
\begin{aligned}
P_f(m) \quad = \quad & \beta_{0f} + \beta_{pf}p(m) + \beta_{df}d(m) + \beta_{bf}b(m) + \sum_{g \in \{g(m)\}}\beta_g + \\
& + \delta_{wf}W_{y(m)} + \delta_{gf}G_{y(m)} + \delta_{nf}N_{y(m)} + \delta_{cf}C_{y(m)} + \\
& + \epsilon_{y(m)f},
\end{aligned} \tag{1}
$$

where $P_f$ is the female representation for movie-making function $f$ in movie $m$ released in year $y$, $p(m)$ and $d(m)$ are the percentage of females in the producing and directing teams, respectively, $b(m)$ is the logarithm of the movie's budget, $\{g(m)\}$ is the movie's set of genres, $W_{y(m)}$ is the female workforce participation, $G_{y(m)}$ is the real GDP per capita change, $N_{y(m)}$ is the number of movies released, $C_{y(m)}$ is the industry concentration, which we measured using the Herfindhal-Hirschman index, the $\beta$s and $\delta$s are the coefficients we aim to estimate through regression, and the $\epsilon$s are assumed to be Gaussian distributed independent noise terms with zero mean (Fig 3). Because the vast majority of movies have a single director, a single cinematographer, and (prior to 1950) a single producer, we perform logistic regression of the model above for those functions. Naturally, when modeling female representation in directing, we do not include the $d(m)$ term, and when modeling female representation in producing, we do not include the $p(m)$ and $d(m)$ terms.

For movies released prior to 1950, we lack budget information for the vast majority of movies. Thus, we perform fits of the model separately for time periods pre- and post-1950. For fits addressing the period prior to 1950, we do not include the term $b(m)$ in the model. In order to test the robustness of our model and of the coefficient estimates, we fit the models for several different time periods. Moreover, and because our data extends over a long time period, we repeat all fits with the inclusion of time dummies. Because the large number of time dummies creates convergence issues, especially for the logistic fits, we consider both yearly (S2 and S3 Figs) and 4-year time dummies (S4 and S5 Figs). Re-assuringly, we find that the model fits for the two periods yield remarkably similar coefficient estimates and that the coefficient for the vast majority of the time dummies are not significantly different from zero.

As suggested by the data in Fig 1, for the period prior to 1950, we find strong, consistent, statistically significant associations between female representation across all movie functions and industry concentration. For the post-1950 period, we again we find strong, consistent, statistically significant associations between female representation in casts and industry concentration, but not for other movie functions. For screenwriters, the post-1950 period coefficient would have been statistically significant at the 0.05 level. However, for directors and cinematographers, the fact that we are conducting a logistic regression and the narrower variation range of industry concentration post-1950, mean that we lack statistical power.

Surprisingly, our analyses do not uncover consistent or significant associations between female representation and national economic indicators. However, confirming the expectations, we find strong, consistent, statistically significant associations between female representation in casts and movie genre. Action, Adventure, Crime, Thriller, and Western have lower female representation, whereas Comedy, Drama, and Romance have higher female

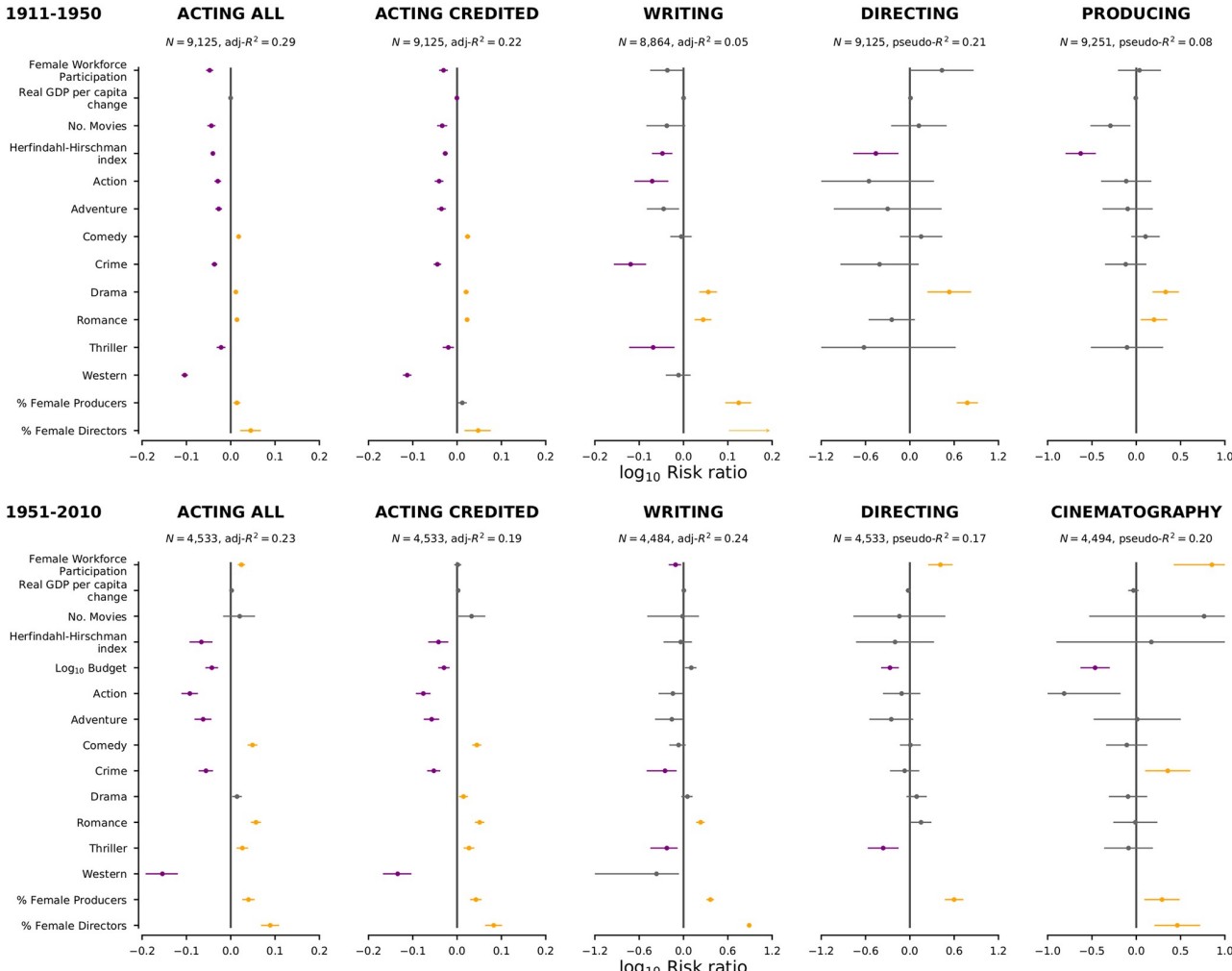

**Fig 3. Multivariate logistic (for director, cinematographer, and producers) and OLS (for actors and writers) regressions models of female representation reveal associations with percentage female producers and genre.** We do not include time dummies in the models for which we show regression results here (see S2–S5 Figs for other time periods and for models including time dummies). We plot the base 10 logarithm of the risk ratio for each factor. The risk factor for female workforce participation was calculated for an increase of 10%. The risk factor for change in real GPD per capita was calculated for an increase of 1%. The risk factor for number of movies was calculated for an increase of 300. The risk factor for industry concentration was calculate for a change of 0.03. The risk factor for budget was calculated for an increase by 100 fold. The risk ratio for percentage female producers was calculated for an increase of 50%. This value is one of the four typical values observed, the other three being 0%, 25% and 100%. The risk factor for percentage female directors was calculated for an increase of 100% since typically there is a single director. Risk ratios for which we are not able to reject the null hypothesis at the 0.01 level are shown in grey. Risk ratios significantly larger than 1 are shown in yellow and those significantly smaller than 1 are shown in purple.

representation. While some of these associations are also consistently present for screenwriters, the vast majority cannot be observed consistently for directors, cinematographers, or producers.

Confirming expectations, we find a significant negative association between movie budget and female representation. Interestingly, the association is especially strong for director and cinematographer. Remarkably, we also find a strong, consistent, and statistically significant positive association between a female director and female representation in cast and screenwriting teams. Our most important result, however, is the consistent and statistically

significant positive association between percentage of female producers and female representation for both periods.

Our regression analyses suggest that females producers and directors contribute to the career advancement of other females in the industry. Our findings thus contrast with survey results [49] suggesting that in many industries females leaders are 'cogs in the machine' and unable to effectuate change in the workplace because they are [49] "either not powerful enough to affect the careers of their subordinates, or they have been selected to their managerial positions *because* they identify with powerful men at the apex of firms". Our findings also contrast with evidence in science hiring for a preference by both males and females for male applicants. [50]

### Changes in producer team characteristics

The pre-1950 period considered in our analyses is particularly interesting because of the dramatic changes in industry concentration. In order to further determine the extent to which the association between industry consolidation and decreased female representation is tied together, we next focus on the period 1918-1930 and calculate female representation for three movie-making functions according to producing company. Specifically, we track the movies produced by the seven companies that would come to dominate the industry during the Studio System: 20th Century, Columbia, Fox, MGM, RKO, Universal, and Warner. We find that prior to 1930, these studios were already producing movies with lower representation of females in acting and producing than the rest of the industry (Fig 4). Thus, it appears quite plausible that the consolidation of the industry centered around the "Big Seven" studios promoted a culture where contributions from females were not welcomed.

The *status quo* of the U.S. movie industry started to change dramatically around 1950 because of the lost legal battles of the 1940s (Fig 1) and the impact of the adoption of television by U.S. households. [47] As a reaction to these changes, the major Hollywood studios focused to a greater extent on big budget movies that made use of technological innovations, such as widescreens and Technicolor, and thus would have a better chance of bringing people to the theaters. [51, 52] The changed focus of the major studios created opportunities for independent competitors [53, 54] and, together with the need for consultants with the desired technical expertise, resulted in a dramatic increase in the number of new producers entering the industry and the size of the producer teams (Fig 5a–5c).

Fig 5 shows how producer team characteristics changed dramatically around this time. Until 1950, a single producer was associated with a movie. From then on, the average number of producers keeps growing and reaches an average of eight by 2010. This pattern is quite distinct from the more abrupt, almost step-like, changes in cast size, or the unchanged number of directors per movie. The changes in the size of producer teams after 1940 look visually similar to the increases in female representation in other movie-making functions. We thus hypothesize that changes in gender representation may have been, at least partially, driven by movie stars freed from the Studio System and with greater negotiation power who were newly able to make production decisions and to change the *status quo*.

To test this possibility, we calculate the fraction of new producers entering the industry who had prior acting experience (Fig 5d). The patterns in the data are consistent with our hypothesis. A first peak, centered at 1922, shows that about a quarter of new producers entering the industry had prior acting credits. This peak coincide with peaks in levels of female representation in casts, directing, and producing. A second and third peaks, centered in 1948 and 1959, respectively, again increased rates of new producers with prior acting credits. Suggestively, these peaks coincide almost exactly with "step jumps" in the probability of directors and producers being female (Fig 1d).

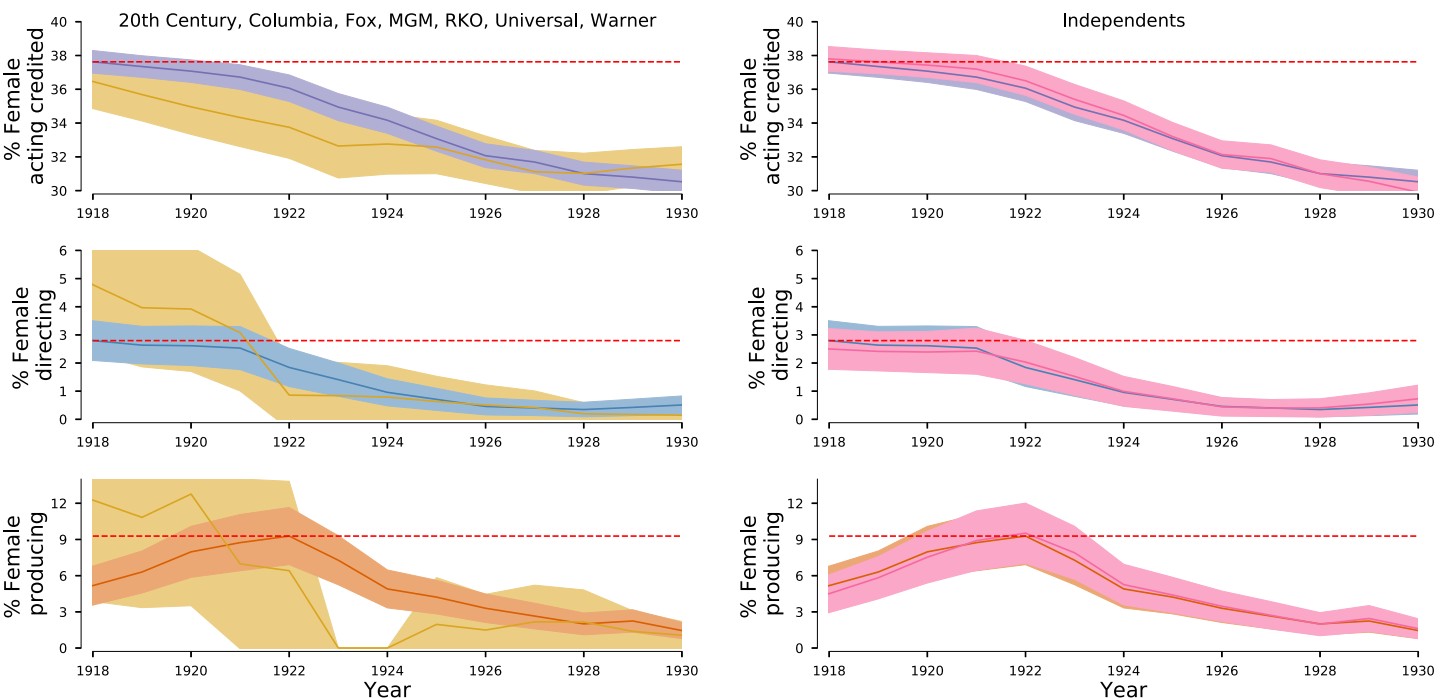

**Fig 4. Gender balance for four movie-making functions produced by the "Big Seven"—we drop disney from the "Big Eight" because during this period they focused on animation movies—and by the so-called Independents.** We compare female representation for all movies and for movies produced by each group of studios. Values for all movies use the same colors as in Fig 1. Values for movies produced by the "Big Seven" are shown in light yellow, and produced by the Independents in pink. Strikingly, during the first half of the 1920s, the "Big Seven" released movies with lower representation levels of females in both acting and producing.

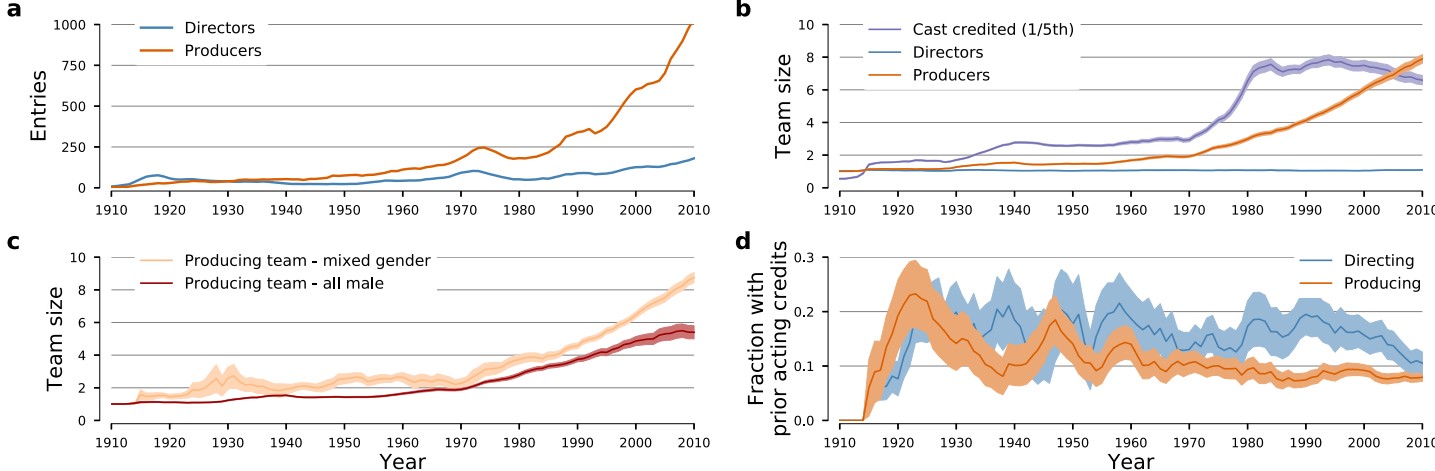

**Fig 5. Entry of new producers and directors into the U.S. movie industry.** (**a**) Number of new producers and new directors entering the industry annually. Except for the 1910s, the number of new producers has systematically exceeded the number of new director entering the industry. (**b**) Temporal evolution of producing team size, directing team size, and cast size. U.S.-produced movies have maintained the tradition of having a single director. In contrast, producing teams have grown steadily in size, from a mean size of one until the mid-1940s to a mean of eight. Interestingly, cast size has grown in a more discontinuous manner, with three step changes occurring around 1915, 1935, and 1975. (**c**) Time dependence of the average size of the team of producers for all male and mixed-gender teams. Color bands show 95% confidence intervals. It is visually apparent, that teams that include females are typically larger and have grown faster. (**d**) Fraction of new producers (directors) who had at least 4 years of prior acting experience at the time of their first producer (director) credit. For directors, that fraction remained approximately constant during the entire period. In contrast, for producers, there are three well-defined peaks centered at 1922, 1948, and 1959. The presence and location of these peaks does not differ if we change the requirement on the duration of the prior acting experience to 6 or 8 years, or if we require a minimum of 5 or 10 prior acting credits.

## Limitations

Our study suffers from three major limitations. First, it does not capture changes in gender representation among starring, as opposed to credited, actors. This limitation is difficult to address in a systematic manner since we do not have a reliable method to determine who within a cast can be classified as a star. This limitation may, however, not be particularly relevant to the detection of systemic issues since stars may be comparatively shielded.

Second, our analysis is restricted to about 26 thousand U.S.-produced movies that are catalogued in the AFI database. Thus, we cannot determine whether similar or distinct trends arose in the movie industry of other countries or in TV scripted shows. However, we believe that at this point it is more important to focus on a less broad case but one for which coverage is comprehensive. [55]

Third, our study can only reveal associations and cannot demonstrate causality. Nonetheless, our analysis support both recent and old accounts of the power held by producers, and the gender bias in awards [56, 57] or collaborations. [58, 59] As demonstrated by the data in Figs 1 and 4, the heydays of the Hollywood Studio System coincided with a disappearance of females from functions with decision-making power. The economic shift in favor of movie stars and the power that shift brought to some actresses enabled them to later play roles as producers and directors coinciding with a virtuous cycle of increased female representation in the industry.

## Discussion

While specific to the movie industry, our finding have broad implications that extend far beyond it. Our results are remarkably consistent with the predictions of the status incongruity hypothesis of Rudman *et al.* [32] Specifically, as an industry grows in importance, there are increasing financial rewards and industry consolidation. This transition may then result in hostility toward successful females and people of color, and create the conditions for a backlash.

In the movie industry, the prominence of the Hollywood studies post World War I, may have made it "unacceptable" to have women and people of color in positions of leadership outside of acting. The hypothetic resulting backlash may then be responsible for the complete diversity collapse observed during the Studio System's Golden Age especially within producing and directing. The fact that women were the most affected by this transformation, may be the reason why two females—actresses Bette Davis, in the 1930s, and Olivia de Havilland, in the 1940s—were the ones to take legal action against the Studio's absolute control over actors' careers.

Our findings can thus be added to the cases of computer science, [8, 9] medicine, [60] and literature, [61] where an increase in the importance of the field was followed by a collapse of diversity. Our findings and the status incongruity hypothesis are also consistent with the loss of prestige of professions, such as elementary school teaching [62] and some medical specialties, [63] that have experienced increases in female representation.

Our study also sheds light on the *plausibility* of some explanations for why females and other groups are under-represented or even absent from some professions. For some careers, it has been argued that males may be superiorly suited because of greater "physical strength," greater "mathematical ability", or some other advantage, or because, unlike females, they do not have to interrupt their careers due to childbearing. In some context, the lack of females has also been attributed to "a shortage of qualified female job candidates." For the case of motion pictures, none of these arguments is plausible. Surveys show that high-school aged girls are more interested in careers in the arts, including acting, than their male peers. [26] Moreover,

there is no credible reason for believing that there are different innate acting abilities and so it cannot be seriously argued that such differences have been the driver of the gender imbalance we report.

Finally, our study provides support for the positive impact of higher female representation at the *decision-making* level in the movie industry. This is remarkable because a number of studies suggest this is not always the case. [49, 50] For example, some research suggests that in many industries females leaders are 'cogs in the machine' and unable to effectuate change in the workplace because they are [49] "either not powerful enough to affect the careers of their subordinates, or they have been selected to their managerial positions *because* they identify with powerful men at the apex".

## Methods

**The hollywood studio system.**   The birth and expansion of the U.S. movie industry coincided with a period of major historical changes (Fig 1a). Movie making started in the U.S. as soon as the technology became available. The U.S.'s insulation from the effects of the first World War, which greatly hindered the development of the European movie industry, enabled the U.S. movie industry to expand rapidly (Fig 1b). By the 1920s, Hollywood was the dominant player in the global movie industry both in terms of the number of movies being produced and in terms of the profits captured [48].

Prior to 1915, the broad expansion of the U.S. movie industry was hindered by the efforts of the Motion Picture Patents Company (MPPC), which represented the major intellectual property owners. The MPPC activities helped increase industry concentration (Fig 1c). As a result, by 1917, the top 5 (out of 97) companies accounted for 38% of U.S. released movies. Immediately after the MPCC dissolution, the degree of concentration of the industry decreased dramatically—by 1920, the top 5 (out of 112) companies accounted for 28% of releases. This diversification trend, however, was reversed with the consolidation of the industry in Hollywood and the establishment of the Studio System. By 1940, 51% of U.S. movies were released by the top 5 (out of 58) companies. While there is a vigorous discussion on the dates for the establishment and consolidation of the Studio System, our analysis of industry concentration strongly suggests that the Studio System started forming around 1920, and remained stable until the mid-1940s.

**Selection criteria for study inclusion.**   We extracted available information for 35,281 U. S.-produced movies compiled by the American Film Institute (AFI, https://search.proquest. com/afi/) and released between 1910 and 2010. We stored, when available, release year, title, and genre. Additionally, we stored credited and uncredited directors, producers, writers, cinematographers, casting directors, cast members, and production companies for each movie. After excluding movies not in English, we were left with 30,706 movies for consideration (S1 File).

Because the AFI database does not provide disambiguated information on industry participants, we also retrieved data from the IMDb which disambiguates industry participants. Specifically, we retrieved, when available, release year, title, genre, budget, revenue, whether it was a made-for-TV movie, credited and uncredited directors, producers, writers, cinematographers, casting directors, and cast members for 393,575 U.S.-produced movies. We also retrieved the IMDb record for the 1,856,569 industry participants identified in these movies. These participants include humans, but also animals and groups (such as orchestras). For each disambiguated individual record, we stored, when available, name, gender, birth year, and death year. For details on gender assignment, see below. Information on the career of industry participants is available at url https://doi.org/10.21985/n2-ttv2-qz74.

We then used titles, years of release and director names, to match AFI to IMDb records. We excluded from the match Made-for-TV movies, and movies classified as Animation or News. Because release year in AFI can be incorrect—that is, stored year of release does not match information on dates of first screening—we allowed for a ±1 difference in year of release between AFI and IMDb records. This matching process identified 26,082 movies (S2 File). As a test of the accuracy of our process, we manually checked 101 randomly selected movies. We found zero matching errors, 7 cases of different release years, and 3 cases for which AFI did not include director information but IMDb did. Because data for 1910 was dominated by Shorts, we restricted the analysis to movies released in the 100 year period 1911–2010. The reduced our final sample to 25,645 movies.

For the 101 matched movies, we also matched cast lists, credited and all, between AFI and IMDb records (S3 and S4 Files). Matching credited casts is challenging because neither database uses consistent criteria for what makes a cast member credited. For this reason, we looked for consensus in the classifications from the two databases. For the credited (all) casts of the randomly selected 101 movies, we matched 1,161 (2035) actors; 2 (51) actors were listed in AFI but not IMDb, and 55 (831) were listed in IMDb but not AFI. After looking in detail at disagreements, we concluded that IMDb cast lists are at least as accurate as AFI's, and most times are more detailed.

**Assigning gender to individuals.**    The gender of actors is explicitly mentioned in most of their individual biographical pages, thus we are able to determine their gender. For individuals with other movie-making functions who do not also have acting credits, we used four independent indirect methods for assigning gender and then build a consensus based on the different assignments.

Method 1: We used the method *get_gender* from the Python package *gender-guesser.detector* (version 0.4.0) to "guess" the gender using the individual's given (first) name. [64] The output of this method is one of 'female,' 'mostly female,' 'androgynous,' 'unknown,' 'mostly male,' or 'male.' Method 2: If present, we parsed an individual's biographical page (http://www.imdb.com/name/person_id/bio) for gender-specific pronouns (he/his/him/himself, or she/her/hers/herself). When the number of (male-) female-specific pronouns exceeded that of (female-) male-specific ones, we assume the individual is a (male) female. Ties are marked as 'unknown.' Method 3: We parsed an individual's biographical page (http://www.imdb.com/name/person_id) for gender-specific pronouns (he/his/him/himself, or she/her/hers/herself). When the number of (male-) female-specific pronouns exceeded that of (female-) male-specific ones, we assume the individual is a (male) female. Ties are marked as 'unknown.' Method 4: We scan the webpage http://www.imdb.com/name/person_id for the html tags with id 'filmo-head-actress' and 'filmo-head-actor.' It assigns the corresponding gender if only one of these searches is successful, otherwise it returns 'unknown.'

Given the results of the four methods, we make a consensus determination using the following algorithm. Each gender assignment of 'male' is worth one point for 'male' gender, each assignment of 'mostly male' is worth a half-point for 'male' gender, and similarly for 'female.' All other assignments are worth no points. The gender with the most points is assigned to the individual. In case of a tie, the gender is assigned as 'unknown.'

**Regression analysis.**    We systematically tested the robustness of our regression analysis. We separate the analysis into two main time periods, pre- and post-1950. This separation is driven by the fact that budget information only becomes available reliably after 1950. For each of these major time period, we repeat the regression fit both with and without time dummies and for different sub-periods. In the main text, we assign a confidence to our coefficient estimates based on the range of regression cases under which we obtain consistent results.

In the Supporting Information, we first show regression models that do not include time dummies. S2 and S3 Figs show results for the periods 1916–1950 and 1951–2010, respectively. Next, we consider regression models that include yearly time dummies. S4 and S5 Figs show results for the periods 1916–1950 and 1951–2010, respectively. Finally, we consider regression models that include 4-year time dummies. S6 and S7 Figs show results for the periods 1916–1950 and 1951–2010, respectively.

## Supporting information

**S1 Fig. U.S. produced movies considered in our study by genre.**
(PDF)

**S2 Fig. Multivariate logistic (for director and producers) and OLS (for actors and writers) regressions models of female representation reveal associations with percentage female producers and genre.** *We do not include time dummies in our model*. We plot the base 10 logarithm of the risk ratio for each factor. The risk factor for female workforce participation was calculated for an increase of 10%. The risk factor for change in real GPD per capita was calculated for an increase of 1%. The risk factor for number of movies was calculated for an increase of 300. The risk factor for industry concentration was calculate for a change of 0.03. The risk factor for budget was calculated for an increase by 100 fold. The risk ratio for percentage female producers was calculated for an increase of 50%. This value is one of the four typical values observed, the other three being 0%, 25% and 100%. The risk factor for percentage female directors was calculated for an increase of 100% since typically there is a single director. Risk ratios for which we are not able to reject the null hypothesis at the 0.01 level are shown in grey. Risk ratios significantly larger than 1 are shown in yellow and those significantly smaller than 1 are shown in purple. If a risk ratio estimate and its confidence bands fall outside the range displayed, we mark this with an arrow. The color of the arrow falls the same convention as the color of the risk ratio estimates.
(PDF)

**S3 Fig. Multivariate logistic (for director and cinematographer) and OLS (for actors, writers, and producers) regressions models of female representation reveal associations with percentage female producers and genre.** We do not include time dummies in our model.
(PDF)

**S4 Fig. Multivariate logistic (for director) and OLS (for actors and writers) regressions models of female representation reveal associations with percentage female producers and genre.** *We consider 1-year time dummies in this case*. Note that for directors the maximum number of iterations (35) was exceeded before the convergence criterion was reached. This is due to the time dummies whose estimation uncertainty is enormous and destabilizes the fit. We do not show results for cinematographers because the time dummies make the matrix singular.
(PDF)

**S5 Fig. Multivariate logistic (for director and cinematographer) and OLS (for actors and writers) regressions models of female representation reveal associations with percentage female producers and genre.** *We consider 1-year time dummies in this case*. Note that for directors the maximum number of iterations (35) was exceeded before the convergence criterion was reached. This is due to the time dummies coefficients whose estimation uncertainty is enormous and destabilizes the fit. We do not show results for cinematographers because the time dummies make the matrix singular.
(PDF)

**S6 Fig. Multivariate logistic (for director and producers) and OLS (for actors and writers) regressions models of female representation reveal associations with percentage female producers and genre.** We consider 4-year time dummies in this case.
(PDF)

**S7 Fig. Multivariate logistic (for director and cinematographer) and OLS (for actors and writers) regressions models of female representation reveal associations with percentage female producers and genre.** We consider 4-year time dummies in this case.
(PDF)

**S1 File. Filtered movies from AFI database.**
(XLSX)

**S2 File. List of movies matched between AFI and IMDb.**
(PDF)

**S3 File. List of 101 randomly selected movies used to verify the degree of agreement between credited cast lists for AFI and IMDb records of matched movies.**
(XLSX)

**S4 File. List of 101 randomly selected movies used to verify the degree of agreement between cast lists for AFI and IMDb records of matched movies.**
(XLSX)

## Acknowledgments

We are in debt to Scott Curtis (Dept. of Radio/Television/Film, Northwestern University, and Communication Program, NU-Qatar) for many insightful comments and suggestions. We are also grateful to Martin Gerlach, Adam Pah, and Thomas Stoeger for stimulating discussions.

## Author Contributions

**Conceptualization:** Luís A. Nunes Amaral, João A. G. Moreira, Murielle L. Dunand.

**Data curation:** Luís A. Nunes Amaral, João A. G. Moreira, Heliodoro Tejedor Navarro.

**Formal analysis:** Luís A. Nunes Amaral, João A. G. Moreira, Murielle L. Dunand.

**Funding acquisition:** Luís A. Nunes Amaral, João A. G. Moreira.

**Investigation:** Luís A. Nunes Amaral, João A. G. Moreira, Murielle L. Dunand.

**Methodology:** Luís A. Nunes Amaral, João A. G. Moreira, Hyojun Ada Lee.

**Project administration:** Luís A. Nunes Amaral.

**Software:** Luís A. Nunes Amaral, João A. G. Moreira, Murielle L. Dunand, Heliodoro Tejedor Navarro.

**Supervision:** Luís A. Nunes Amaral, João A. G. Moreira.

**Validation:** Luís A. Nunes Amaral, João A. G. Moreira.

**Visualization:** Luís A. Nunes Amaral, João A. G. Moreira, Murielle L. Dunand.

**Writing – original draft:** Luís A. Nunes Amaral, João A. G. Moreira.

**Writing – review & editing:** Luís A. Nunes Amaral, João A. G. Moreira, Murielle L. Dunand, Hyojun Ada Lee.

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
