## [Decision Letter · Decision Letter 0]

23 Dec 2019

PONE-D-19-30836

Long-term patterns of gender imbalance in an industry without ability or level of interest differences

PLOS ONE

Dear Dr. Amaral,

Thank you for submitting your manuscript to PLOS ONE. After careful consideration, we feel that it has merit but does not fully meet PLOS ONE’s publication criteria as it currently stands. Therefore, we invite you to submit a revised version of the manuscript that addresses the points raised during the review process.

We would appreciate receiving your revised manuscript by Feb 04 2020 11:59PM. To enhance the reproducibility of your results, we recommend that if applicable you deposit your laboratory protocols in protocols.io, where a protocol can be assigned its own identifier (DOI) such that it can be cited independently in the future. For instructions see: http://journals.plos.org/plosone/s/submission-guidelines#loc-laboratory-protocols

We look forward to receiving your revised manuscript.

Kind regards,

Bing Xue, Ph.D.

Academic Editor

PLOS ONE

Journal Requirements:

2. Please upload a copy of Figure 1e-f, to which you refer in your text on page 5. If the figure is no longer to be included as part of the submission please remove all reference to it within the text.

Reviewers' comments:

Reviewer's Responses to Questions

**Comments to the Author**

1. Is the manuscript technically sound, and do the data support the conclusions?

Reviewer #1: Yes

Reviewer #2: Partly

2. Has the statistical analysis been performed appropriately and rigorously? 

Reviewer #1: Yes

Reviewer #2: Yes

3. Have the authors made all data underlying the findings in their manuscript fully available?

Reviewer #1: Yes

Reviewer #2: Yes

4. Is the manuscript presented in an intelligible fashion and written in standard English?

Reviewer #1: Yes

Reviewer #2: Yes

5. Review Comments to the Author

Reviewer #1: 1) well-written ms

2) very interesting data (data collection, analyses, and presentation are actually really good)

3) Big problem with theoretical background: The authors present numerous gender differences (or, as they call it, „gender imbalances“) and kind of like wonder why they exist. Now, the findings they present are (mostly) the ones that any informed evolutionary psychologist would predict. The authors, of cource, do not need "to like" EP, but, as an influential approach that has produced an abundance of evidence-based research in the past decades within the research of gender, the authors need to, at least, acknowledge the existence of that approach - particularly because it offers a very parsimonious explanation of actual gender differences. Of course, not a single approach can explain everything. So, we need a reconcilation of environmental approaches with biological ones. The authors cite Wuhr et al. In that paper, there is a very well-balanced presentation of different approaches that try to explain gender differences in certain domains, including the media. This is how it should be done. Unfortunately, the authors rely - more or less - on social constructivism instead. Granted, they do seem to have found some cultural factors that influence the studied phenomenon. I am fine with that. And I believe that his paper should be published eventually. However, what they found is only a small part of the whole story. It seems, according to the plethora of available (but not cited…) research, as if there is a very strong effect of binary gender (as predictor) on the production of cultural products (as criterion) with environmental effects (the focus of the paper at hand) being („only“) kind of the moderator variable. Apart from gender, there is a predictable and probably quite strong effect of age as well.

4) Referring to 3) I recommend to the authors to have a look at this literature:

The authors talk about computer software. It might be useful to look at Melzer (2018) and Lange & Schwab (2018).

Generally, the pattern of „gender imbalance“ is a pattern that is found across all sorts of cultural products (nota bene: „cultural“ does not necessarily mean „non-biological“):

Literature: Lange, & Euler (2014); Miller (1999)

Music (Jazz): Miller (1999)

Paintings: Miller (1999)

Religions: Lange et al. (2013)

Scientific discoveries: Kanazawa (2000)

World Records: Lange et al. (2013)

I also recommend: Griskevicius et al. (2006); Hennighausen, & Schwab (2015); Kanazawa (2003); Simonton (e.g., 1975)

Generally, Geoffrey Miller has produced numerous works that might be worthwhile to have a look at. Also, many works by Dean K. Simonton in general are extremely valuable.

I do not recommend to switch to a biological approach. Just be more balanced in terms of how you explain your findings … on - what a coincidence! - gender IMBALANCE.

I am entirely with the auhtors in that we need to do all people justice. So, all imbalances pose a problem. But the same can be applied to scientific theories. The paper at hand is a perfect example of an imbalance in scientific argumentation. It is not all about culture, there are also innate stable differences between people.

The authors should also try to be more differentiated. Comparing patterns of movie actors and actresses with the one for movie producers and directors is not a good one. For most movies, you have at least one female and one male lead. So you necessarily need an actress and an actor. (I have to admit though that some movies only have a male lead, which calls for an explanation.) Contrary, the gender of a director is not pre-determined by such necessities. Also, medicine is not well-suited to be compared to the media. There are many different domains within medicine. One motive within medicine is the nursing aspect (with women, ON AVERAGE, being more prone to). Being a heart surgeon, for instance, is a different story though, as it consists more of technical details (with men, ON AVERAGE, being more prone to; see Su et al., 2009 for gender differences in the preference for people in contrast to the preference for things). Also, just try to think about a different causality here. (I know that you state that you cannot demonstrate causality. But there is clearly an implied causality in the paper at hand.) High-quality research shows that in more egalitarian cultures, gender differences are not decreased but increased – which is exactly the opposite of what social constructivism would predict. Quite obviously (see literature), there is a gender difference in motivation to perform well in numerous domains (and not a gender difference in competence!). So, it is not helping to simply state that „different innate […] abilities“ do not exist. Rather, try to think about innate differences in 1) interests and 2) motivation (to achieve high social status). Individual differences in motivation should not be confused with cultural factors.

Please be also more careful with the use of the term „stereotype“ (see Jussim et al., 2009).

To conclude, please do not get me wrong. This is valuable research that should be published. But first, some quite obvious shortcomings need to be adressed.

Literature cited in this review:

Griskevicius, V., Cialdini, R. B., & Kenrick, D. T. (2006). Peacocks, Picasso, and parental investment: The effects of romantic motives on creativity. Journal of Personality and Social Psychology, 91(1), 63-76.

Hennighausen, C. & Schwab, F. (2015). Evolutionary media psychology and ist epistemological foundation. In T. Breyer (Ed.), Epistemological dimensions of evolutionary psychology (pp. 131-158). New York: Springer.

Jussim, L., Cain, T. R., Crawford, J. T., Harber, K., & Cohen, F. (2009). The unbearable accuracy of stereotypes. In T. D. Nelson (Ed.), Handbook of prejudice, stereotyping and discrimination (pp. 199–227). New York, NY: Psychology Press.

Kanazawa, S. (2000). Scientific discoveries as cultural displays: A further test of Miller’s courtship model. Evolution and Human Behavior, 21, 317–321.

Kanazawa, S. (2003). Why productivity fades with age: The crime–genius connection. Journal of Research in Personality, 37(4), 257-272.

Lange, B. P., & Euler, H. A. (2014). Writers have groupies, too: High quality literature production and mating success. Evolutionary Behavioral Sciences, 8(1), 20-30. doi:10.1037/h0097246

Lange, B. P., Schwarz, S. & Euler, H. A. (2013). The sexual nature of human culture. The Evolutionary Review: Art, Science, Culture, 4(1), 76-85.

Lange, B. P., & Schwab, F. (2018). Game on: Sex differences in the production and consumption of video games. In J. Breuer, D. Pietschmann, B. Liebold & B. P. Lange (Eds.), Evolutionary psychology and digital games: Digital hunter-gatherers (pp. 193-204). New York, NY: Routledge.

Melzer, A. (2018). Of princesses, paladins, and players: Gender stereotypes in video games. In J. Breuer, D. Pietschmann, B. Liebold & B. P. Lange (Eds.), Evolutionary psychology and digital games: Digital hunter-gatherers (pp. 205-220). New York, NY: Routledge.

Miller, G. F. (1999). Sexual selection for cultural displays. In R. Dunbar, C. Knight, & C. Power (Eds.), The evolution of culture. An interdisciplinary view (pp. 71-91). Edinburgh: Edinburgh University Press.

Simonton, D. K. (1975). Age and literary creativity: A cross-cultural and transhistorical survey. Journal of Cross-Cultural Psychology, 6, 259–277.

Su, R., Rounds, J., & Armstrong, P. I. (2009). Men and things, women and people: a meta-analysis of sex differences in interests. Psychological Bulletin, 135, 859–884. doi: 10.1037/a0017364

Reviewer #2: I appreciate the opportunity to review your article, entitled “Long-term patterns of gender imbalance in an industry without ability or level of interest differences.” Indeed, I found your paper interesting and provocative. A clear strength of your paper is, quite obviously, the particularly large dataset. Moreover, I liked your approach to examine the representation of women from different angles (e.g., the representation of women in different roles, such as actors and directors; examining different time periods; the additional analysis on the “Big Seven”). That being said, however, I also had several questions and issues that I would like to raise here. I hope these will be helpful as you further advance this work.

1) I found it somewhat confusing that you started by pointing that “one would expect to see parity,” but then go on to suggest that gender parity is not actually expected (e.g., due to discrimination). Thus, my suggestion is to be more precise here and clarify that certain mechanisms that are sometimes discussed to drive gender differences are unlikely to operate in this context. Thus, this context is interesting because certain mechanisms can be examined in greater isolation.

2a) Page 3, lines 39-40: I agree that it is quite difficult to know “why a SPECIFIC individual discriminates” (emphasis added). However, there is abundant research on the reasons why people discriminate more generally. To give just two examples, both role congruity theory (Eagly & Karau, 2002) and the status incongruity hypothesis (Rudman et al., 2012) help to explain why people engage in discrimination. Going forward, I suggest clarifying this issue. This issue also leads me to raise the following issue (i.e., 2b).

2b) Theoretical depth: I understand that you hinted at different processes in your manuscript (e.g., p. 3, line 41) and, most importantly, I do not want to encourage the practice of “hypothesizing after the results are known” (HARKing; Kerr, 1998). However, I would greatly appreciate more theoretical depth and clear rationales that help to illuminate what was found. Going into this direction could help to strengthen the paper and increase its impact. For instance, it would help to elaborate (in the Discussion section) on the questions of why exactly the consolidation of the studio system as well as the percentage of women producers had such an impact. Certain extant theories (see Issue 2a above) likely help to make sense of the findings.

3) Regression models: As noted in your paper (lines 86-87), “the percentage of females in the producing and directing teams” were predictors. Thus, it is obvious that including these predictors in the models to predict the representation of women as directors and producers is problematic, as this would reflect autocorrelations. I assume this is the reason why the respective estimates for "% female producers" and "% female directors" are not included in certain parts of Figure 3 (and also, for instance, Figure S3). Going forward, please clarify whether or not the model was different when predicting the representation of women as directors and producers specifically.

4) I appreciate your robustness checks. Yet, please explain why you examined “time” by including a number of dummy variables, rather than by including time as a continuous variable.

5) Page 7, line 107: I would like to know what leads you to reason that low statistical power is the reason for the absence of the predicted effects. The sample size generally appears quite large.

Once again, I hope that these comments are helpful, as you are clearly onto something interesting.

References:

Eagly, A. H., & Karau, S. J. (2002). Role congruity theory of prejudice toward female leaders. Psychological Review, 109, 573-598.

Kerr, N. L. (1998). HARKing: Hypothesizing after the results are known. Personality and Social Psychology Review, 2, 196-217.

Rudman, L. A., Moss-Racusin, C. A., Phelan, J. E., & Nauts, S. (2012). Status incongruity and backlash effects: Defending the gender hierarchy motivates prejudice against female leaders. Journal of Experimental Social Psychology, 48, 165-179.

6. PLOS authors have the option to publish the peer review history of their article (what does this mean?). If published, this will include your full peer review and any attached files.

Reviewer #1: No

Reviewer #2: No

---

## [Decision Letter · Decision Letter 1]

7 Feb 2020

PONE-D-19-30836R1

Long-term patterns of gender imbalance in an industry without ability or level of interest differences

PLOS ONE

Dear Dr. Amaral,

Thank you for submitting your manuscript to PLOS ONE. After careful consideration, we feel that it has merit but does not fully meet PLOS ONE’s publication criteria as it currently stands. Therefore, we invite you to submit a revised version of the manuscript that addresses the points raised during the review process.

We would appreciate receiving your revised manuscript by Mar 23 2020 11:59PM. To enhance the reproducibility of your results, we recommend that if applicable you deposit your laboratory protocols in protocols.io, where a protocol can be assigned its own identifier (DOI) such that it can be cited independently in the future. For instructions see: http://journals.plos.org/plosone/s/submission-guidelines#loc-laboratory-protocols

We look forward to receiving your revised manuscript.

Kind regards,

Bing Xue, Ph.D.

Academic Editor

PLOS ONE

Reviewers' comments:

Reviewer's Responses to Questions

**Comments to the Author**

1. If the authors have adequately addressed your comments raised in a previous round of review and you feel that this manuscript is now acceptable for publication, you may indicate that here to bypass the “Comments to the Author” section, enter your conflict of interest statement in the “Confidential to Editor” section, and submit your "Accept" recommendation.

Reviewer #1: (No Response)

Reviewer #2: All comments have been addressed

2. Is the manuscript technically sound, and do the data support the conclusions?

Reviewer #1: Yes

Reviewer #2: Yes

3. Has the statistical analysis been performed appropriately and rigorously? 

Reviewer #1: Yes

Reviewer #2: Yes

4. Have the authors made all data underlying the findings in their manuscript fully available?

Reviewer #1: Yes

Reviewer #2: Yes

5. Is the manuscript presented in an intelligible fashion and written in standard English?

Reviewer #1: Yes

Reviewer #2: Yes

6. Review Comments to the Author

Reviewer #1: The authors have reworked their MS considerably. However, sadly, they have mostly ignored (or misunderstood) what I was criticising. They have added Miller (1999) and Lange et al. (2013) – two of many works I had recommended. That is basically good. However, althought they cite some of the suggested works now, R1 of the MS leaves many doubts if they understood what those works are asserting (based on empirical data of course). To make it more concrete, they state (p. 3): „Indeed, there is currently a great deal of controversy surrounding the primacy of evolutionary/biological mechanisms versus environmental/cultural mechanisms in explaining observed gender representation imbalances.“ Correct, nothing to criticise here. They go on: „Even though we pursue here an approach informed by cultural primacy approaches, we believe that our study is able to sidestep the controversy….“ Ok too, although this is were an informed biosocial researcher should get nervous. And indeed, the full sentence goes as follows: „Even though we pursue here an approach informed by cultural primacy approaches, we believe that our study is able to sidestep the controversy because the movie industry provides a case study in which evolutionary mechanisms actually would lead to the expectation that women would be over-represented. Specifically, studies informed by evolutionary psychology, suggest that women have a preference for activities where people and inter-personal interactions are prominent.“ First, how can you sidestep a controvery, while in the very same sentence diving fully into the very same controversy? Secondly and more importantly, the statement is utterly wrong. No informed evolutionary psychologist would predict that! The reference they use for this awkward statement is (no wonder) a non-evolutionary one (Lewis P. and Simpson R. Kanter revisited: Gender, power and (in)visibility. International Journal of Management Reviews, 14:141{158, 2012.). They are so many good references for this topic, and I have invested so much time in my first review mentioning many of them to the authors. This is really disappoiting! I am investing so much time to try to help the authors to end up having a sound piece of science. And this is what happened. �

As I have already explained in my first review: When it comes to gender differences (on average, of course) in the context of „social“ occupations, you need to be more differentiated. The motivation of an average men in the context of „social“ is to strive for social status, that is to climb the social ladder in order to get an upper spot in the social hierarchy. This pattern is cross-culturally universal and also in terms of time very stable. So the most parsimonious explanation for this would be to assume that this pattern is partially due to evolutionary selection pressures. How exactly? Men can benefit from high social status in terms of so-called fitness benefits over-proportionally compared to women of high social status. On the contrary, „social“ in the context of occupations that women are, on average, more interested in is kind of synonymous with nurturing and caring. The authors themselves state: „women have a preference for activities where people and inter-personal interactions are prominent“.

I won’t go into more detail, because (1) I have explained this in my first review already and (2) I have suggested so many works that explain it in detail and support the assertion by strony empirical evidence.

I leave it to the authors to change this part of their paper in order to make it correct or not.

Reviewer #2: I appreciate the opportunity to review a revised version of your manuscript. You have addressed all of my comments thoroughly: My methodological questions were clarified. Moreover, I greatly appreciate the greater theoretical depth of this revision. This way, the potential impact of your paper might be even stronger.

7. PLOS authors have the option to publish the peer review history of their article (what does this mean?). If published, this will include your full peer review and any attached files.

Reviewer #1: No

Reviewer #2: No

---

## [Editor Report · Decision Letter 2]

12 Feb 2020

Long-term patterns of gender imbalance in an industry without ability or level of interest differences

PONE-D-19-30836R2

Dear Dr. Amaral,

We are pleased to inform you that your manuscript has been judged scientifically suitable for publication and will be formally accepted for publication once it complies with all outstanding technical requirements.

With kind regards,

Bing Xue, Ph.D.

Academic Editor

PLOS ONE
---

## [Editor Report · Acceptance letter]

28 Feb 2020

PONE-D-19-30836R2 

Long-term patterns of gender imbalance in an industry without ability or level of interest differences 

Dear Dr. Amaral:

I am pleased to inform you that your manuscript has been deemed suitable for publication in PLOS ONE. Congratulations! Your manuscript is now with our production department. 

With kind regards,

on behalf of

Professor Bing Xue 

Academic Editor

PLOS ONE